# One-to-Normal: Anomaly Personalization for Few-shot Anomaly Detection

**Yiyue Li**[1]    **Shaoting Zhang**[4]    **Kang Li**[134]*    **Qicheng Lao**[24]*

[1]West China Biomedical Big Data Center, West China Hospital, Sichuan University
[2]School of Artificial Intelligence, Beijing University of Posts and Telecommunications
[3]Sichuan University Pittsburgh Institute, Sichuan University
[4]Shanghai Artificial Intelligence Laboratory
`kang.li.research@gmail.com, qicheng.lao@bupt.edu.cn`

## Abstract

Traditional Anomaly Detection (AD) methods have predominantly relied on unsupervised learning from extensive normal data. Recent AD methods have evolved with the advent of large pre-trained vision-language models, enhancing few-shot anomaly detection capabilities. However, these latest AD methods still exhibit limitations in accuracy improvement. One contributing factor is their direct comparison of a query image's features with those of few-shot normal images. This direct comparison often leads to a loss of precision and complicates the extension of these techniques to more complex domains—an area that remains underexplored in a more refined and comprehensive manner. To address these limitations, we introduce the anomaly personalization method, which performs a personalized one-to-normal transformation of query images using an anomaly-free customized generation model, ensuring close alignment with the normal manifold. Moreover, to further enhance the stability and robustness of prediction results, we propose a triplet contrastive anomaly inference strategy, which incorporates a comprehensive comparison between the query and generated anomaly-free data pool and prompt information. Extensive evaluations across eleven datasets in three domains demonstrate our model's effectiveness compared to the latest AD methods. Additionally, our method has been proven to transfer flexibly to other AD methods, with the generated image data effectively improving the performance of other AD methods.

## 1 Introduction

Anomaly Detection (AD) has garnered considerable attention due to its wide applicability across various domains, such as industrial defect detection [20, 30, 2, 14, 27], medical diagnostics [37, 21, 43], video surveillance [22, 33], manufacturing inspection [39, 9]. This heightened interest is primarily attributed to its only reliance on normal samples and its adoption of an unsupervised learning paradigm, where it typically learns from the distribution of normal samples to identify anomalies by detecting outliers [46, 39]. Traditional AD methods, including auto-encoder based [43, 40], GAN-based, and knowledge-based approaches [3, 8, 35], and others. Moreover, some diffusion-based methods [24, 11] have also emerged recently. Although most of these methods do not require annotated data, they do necessitate a substantial number of normal samples during the training phase to capture the distribution of normal samples effectively. However, this requirement has also severely constrained its advancement in various fields.

---

* Corresponding author

Recent studies [36, 14, 38] in few-shot scenarios have improved AD tasks. Currently, state-of-the-art (SOTA) advancements [45, 16, 10, 44] are primarily due to the development of large pre-trained Visual-Language Models (VLMs). For example, WinCLIP [16] first uses pre-trained CLIP, employing carefully designed text prompts and image feature comparisons to perform few-shot anomaly detection. AnomalyGPT [10] eliminates the need for manually setting thresholds and supports multi-round dialogues. InCTRL [45] achieves general anomaly detection through in-context residual learning. Despite these innovations, the latest SOTA methods often rely on direct feature matching between few-shot normal images (i.e., reference images) and the query sample. However, without deeply exploring their subtle features, it is often difficult to achieve precise feature comparisons, which can easily lead to unstable results, since the non-anomalous differences between query and references can severely impact the prediction accuracy. Furthermore, the limited number of normal images serving as independent references also makes their prediction insufficient.

In this work, we hypothesize that: 1) To achieve more accurate prediction results, it is essential to compare the query image with its corresponding or most similar normal image. Ideally, this comparison should involve a one-to-one transformation of the query image into its normal counterpart; 2) Additionally, one should also aim for robust and stable results, necessitating a comprehensive approach to accurately predicting results from multiple perspectives. To this end, we propose an anomaly personalization method for few-shot anomaly detection. To achieve more accurate, personalized comparative predictions, as mentioned in 1), our method employs a diffusion model to create an anomaly-free customized model inspired by [31]. This model uses pairs of object text prompts and few-shot normal images (i.e., reference images) to explore the distribution of normal samples. Previous works [23, 4, 41] have demonstrated the effectiveness of such customized models. Furthermore, the suitability of diffusion models stems from their powerful customization capabilities, allowing for flexible control over intermediate steps and the generation process compared to other methods [26]. Additionally, for enhanced stability and robustness as mentioned in 2), we propose the use of triplet contrastive anomaly inference, i.e., in addition to comparing the query image with normal samples, we conduct comprehensive comparisons with personalized samples and text prompts. Furthermore, we use comprehensive state words and templates as text prompts inspired by [15, 16]. Ultimately, we synthesize predictions from diverse perspectives to yield the final anomaly score.

To summarize, this paper makes the following main contributions:

- We introduce a novel anomaly personalization method for few-shot anomaly detection, unlike other state-of-the-art (SOTA) approaches that directly compare query images with reference images, our method enables a finer-grained comparison through one-to-normal personalization of query images, leading to enhanced prediction accuracy.

- To achieve more stable and robust results, we propose a triplet contrastive anomaly inference strategy. This approach facilitates a more comprehensive comparison by incorporating not only customized comparisons but also comparisons with anomaly-free samples and text prompts. The anomaly-free samples augment normal images sampled by an anomaly-free customized model.

- Our proposed method demonstrates strong generalizability. We conduct comprehensive experiments across 11 datasets spanning three distinct domains: industrial, medical, and semantic. Moreover, the anomaly-free samples generated by our method can be used to augment the normal samples in most few-shot anomaly detection methods, enhancing the performance of some existing methods and demonstrating adaptability and flexibility.

## 2 Related Work

### 2.1 Anomaly Detection

Anomaly detection (AD) is a critical task in computer vision aimed at identifying samples that deviate significantly from the norm. Traditional AD methods can be categorized into several types: auto-encoder based [43, 40], GAN-based, and knowledge-based approaches [3, 8, 35], and others. Recent advancements in AD research focus on few-shot and zero-shot learning to overcome data limitations. Few-shot AD methods like RegAD [14] utilize pre-trained models and a few normal samples from the target domain to detect anomalies without extensive re-training. WinCLIP [16] meticulously designs a variety of prompts to ensure the model comprehensively covers all possible normal and abnormal scenarios. AnomalyCLIP [44] uses CLIP for zero-shot anomaly detection.

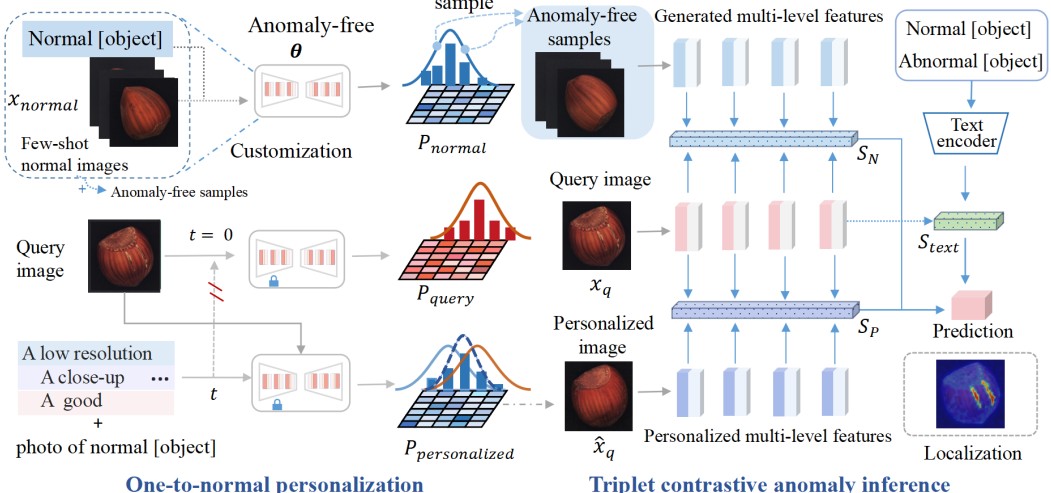

**One-to-normal personalization**      **Triplet contrastive anomaly inference**

Figure 1: Overview of our proposed anomaly personalization approach. First, we use few-shot normal images to customize an anomaly-free diffusion model (*top left*), similar to Dreambooth. Next, we perform one-to-normal personalization of the query image, obtaining the personalized image (*bottom left*). Finally, we adopt a triplet contrastive anomaly inference method, which integrates anomaly scores from three aspects to obtain the final results. $S_N$, $S_P$ and $S_{text}$ represent different contrastive anomaly inference processes.

They have also meticulously designed prompts but have overlooked the aspect of image comparison. In summary, many of the current few-shot AD approaches have primarily focused on text prompt refinement without detailed exploration in visual feature comparisons and sample generation.

## 2.2 Image Personalization

Personalized text-to-image generation has emerged as a pivotal area, particularly for creating highly personalized and contextually accurate images. DreamBooth [31] is the first to focus on fine-tuning diffusion models with specific subject images to generate high-fidelity renditions of those subjects, typically using 3 to 5 images for customization and incorporating prior preservation to prevent language draft. SuTi [4] leverages a single model to learn from a multitude of expert models, each fine-tuned to specific subjects, allowing for instant personalization using in-context learning with only a few examples. Another innovative approach [23] involves "concept neurons" in diffusion models, identifying clusters of neurons that correspond to specific subjects within a pre-trained model. These methods have achieved significant results in image generation. However, most of these methods have been applied to image editing and synthesis, and have yet to be explored for anomaly detection.

## 3 Method

### 3.1 Overview

We propose a novel anomaly personalization method to improve few-shot anomaly detection. First, to obtain more precise results, we aim for a one-to-one customized comparison, where the query image is compared with the image derived from itself, transformed towards an anomaly-free distribution to align the anomalous distribution with the normal state. This is achieved by employing an anomaly-free customized model (Sec. 3.2) followed by one-to-normal personalization of the query image (Sec. 3.3). Second, to enhance the detection stability and robustness, we further propose a triplet contrastive anomaly inference process (Sec. 3.4), where the query image is thoroughly compared with its personalized version, augmented anomaly-free samples, and text prompts for comprehensive analysis.

## 3.2 Anomaly-Free Customized Model

To develop a customized model for normal objects, we choose diffusion model [12] as the framework, building on the work based on [31] which has proven effective for specific object customization. To enhance the diversity of normal object samples given limited data, we first apply data augmentation to normal reference images. Subsequently, to better align objects with their textual descriptions, we provide the diffusion model with a series of demonstrations $\mathbb{C}_o$, using augmented reference images paired with their corresponding texts for specific objects. Specifically, for each object (e.g, cable), we prepare a series of augmented reference images (i.e., few-shot normal samples) $x_{\text{normal}} \in \mathcal{D}_{\text{train}}$ and their matching text $c$, i.e., *a photo of normal [object]*, for tailored training of the diffusion model. Here, the specific category name (e.g., 'cable') is used to replace 'object'. Additionally, another set of demonstrations $\overline{\mathbb{C}_o}$ is employed to design a class-specific prior preservation loss, aimed at promoting diversity and preventing language drift, similar to [31, 4]. To obtain an anomaly-free customized model $D_\theta(x_{\text{normal},t}, c)$ parameterized by $\theta$ on an object $o$, we constrain a pre-trained diffusion model on the image cluster $\mathbb{C}_o$ with the denoising loss as:

$$\theta = \arg\min_\theta \mathbb{E}_{(x_0,c)\sim\mathbb{C}_o\cup\overline{\mathbb{C}_o}}\{\mathbb{E}_{\epsilon,t}[||D_\theta(x_{\text{normal},t}, c) - x_0||_2^2\}, \tag{1}$$

where $c$ is a short description of normal images $x_{\text{normal}}$, and $x_{\text{normal},t}$ is a latent version of the input noised by $t$ steps. Note that the dimensionality of both the image and the latent is the same throughout the entire process. Therefore, the anomaly-free customized model learns the distribution of normal images $P_{\text{normal}}$ conditioned on the text prompt $c$, where anomaly-free samples can be generated.

## 3.3 One-to-Normal Personalization

Our proposed one-to-normal strategy is designed to transform the query image of an object toward the distribution of its normal samples. It adaptively retains the normal characteristics while the anomaly parts are gradually transformed with no defects. This is similar to previous work [23, 26] for image editing; however, our strategy specifically focuses on utilizing it to align with the normal manifold.

Specifically, we first design text prompts to maximize the retention of information in the normal regions of the query image while transforming anomalous areas towards a normal state. To mitigate the influence of other factors (e.g., contrast, image quality), we simulate all potential normal states using these prompts. Inspired by [16, 44], we curate a template list for various potential image physical states, e.g., *a low-resolution photo of a normal object*. Additionally, we include descriptors for common states shared by most normal objects, such as *a photo of normal object without flaw*. We provide these descriptors for normal state prompts in Appendix A.1.

Given a series of normal state prompts $\{c\}_{i=1}^n$ described above, we aim to select the generated image that most closely resembles the normal state for the one-to-normal personalization of the query image $x_q$. Specifically, we reconstruct a set of reconstructed images $\{\hat{x}_i\}_{i=1}^n$ where each text prompt $c_i$ guides the diffusion process of the customized anomaly-free model. We select the optimal text prompt $c_q$ that most closely resembles the normal state from the prompt set:

$$c_q = \arg\min_{c_i} \mathcal{L}(x_q, \hat{x}_i) = \arg\min_{c_i} \mathcal{L}(x_q, D_\theta(x_{q,t}, c_i, t)), \tag{2}$$

where $D_\theta$ represents the anomaly-free customized diffusion model derived from the Section 3.2, and $\mathcal{L}$ represents SSIM. Finally, we obtain the personalized images $\hat{x}_q$:

$$\hat{x}_q = D_\theta(\sqrt{\alpha_t}x_q + \sqrt{1-\alpha_t}\epsilon, c_q, t), \tag{3}$$

where $\alpha$ is a noise schedule, $\epsilon \, \mathcal{N}(0, I)$ represents Gaussian noise, and $t$ is the diffusion step.

Following [25], during this noising phase, the hyperparameter $t$ is meticulously adjusted to dictate the extent of the one-to-normal personalization, which determines the similarity between query image $x_q$ and personalized image $\hat{x}_q$. A relatively lower $t$ means insufficient personalization where a substantial portion of the original query is retained. As $t$ approaches 0, the generated image becomes identical to the input query image, and the text prompt's condition has no effect. Conversely, setting $t$ to $T$ initiates a complete forward diffusion process, consequently erasing all information in $x_q$ and

allowing the generation to be fully guided by text prompts $c_q$ toward the normal manifold. Ultimately, we bridge the distributions of $P_{\text{query}}$ and $P_{\text{normal}}$ for obtaining $P_{\text{personalized}}$, from which we can sample personalized images for subsequent anomaly detection tasks.

## 3.4 Triplet Contrastive Anomaly Inference

To achieve comprehensive predictions and enhance the precision and robustness, we introduce triplet contrastive anomaly inference, i.e., in addition to comparing the query image with anomaly-free samples and text prompts, we also compare it with our personalized images. Finally, we integrate predictions from three comparison aspects to mutually complement each other.

**One-to-one personalized comparison.** We divide the CLIP image encoder into $n$ multi-feature extraction blocks, each designed to capture the intermediate features of the input image at different levels. For a query image $x_q$ and its corresponding personalized image $\hat{x}_q$, the model extracts features $F_q \in \mathbb{R}^{h \times w \times d}$ from $x_q$ and $\hat{F}_q \in \mathbb{R}^{h \times w \times d}$ from $\hat{x}_q$. The comparison score $S_P$ between the query image and its personalized image is then computed by evaluating their similarity in the extracted features at each level:

$$S_P = \frac{1}{n} \sum_{l=1}^{n} \max_{\mathcal{G}} (1 - \langle F_{q,l}, \hat{F}_{q,l} \rangle), \tag{4}$$

where $l$ denotes the $l$-th level feature extraction block, $n$ represents the total number of blocks, and $\langle \cdot \rangle$ signifies the cosine similarity function, $\mathcal{G}$ represents the grid number.

**Anomaly-free sample comparison.** Our anomaly-free sample pool comprises a set of normal reference images and generated normal images, better representing the distribution of normal samples. We first generate normal samples from the customized anomaly-free model $D_\theta$ in Section 3.2 and incorporate them into the anomaly-free sample pool for subsequent prediction tasks, as previous studies have demonstrated the diffusion model's capability to synthesize high-fidelity images, which have been proven effective in many works [41, 1, 25]. To further expand the anomaly-free sample pool, we use the same process described in Section 3.3 but with settings that more closely align with the distribution of normal reference image instead of the query image. Then, we employ the same CLIP image encoder to extract multi-level features from anomaly-free samples (e.g., 30) and store these features in a memory bank $M$. The prediction score $S_N$ between the query image and anomaly-free samples can be expressed as:

$$S_N = \frac{1}{n} \sum_{l=1}^{n} \max_{\mathcal{G}} (\min_{m \in M} (1 - \langle F_{q,l}, m_l \rangle)). \tag{5}$$

**Text prompt comparison.** To calculate the anomaly score between the query image features and the text prompt features, we categorize text prompts into two types: normal and abnormal objects, similar to [16, 5, 15]. We aim to cover more possible states for these objects to better simulate the various potential conditions of the images, similar to [16]. Specifically, we obtain text features by applying the CLIP text encoder to a set of predefined normal and abnormal prompts and then calculating the average of these features, denoted as $F_{text} \in \mathbb{R}^{2 \times d}$. To facilitate comparison, the global image feature for anomaly detection obtained from the CLIP image encoder is represented as $F_q \in \mathbb{R}^{1 \times d}$. These two features from text prompts and the query image are then used to calculate the anomaly score as follows:

$$S_{text} = softmax(F_q F_{text}^T). \tag{6}$$

Here, the score $S_{text}$ represents the probability corresponding to the anomaly.

The final prediction result is generated by combining outputs from three branches: the personalized image, anomaly-free samples, and text prompts. These three sets of output scores provide complementary information for collaboration. The final anomaly score can be obtained by combining the three branches:

$$\mathcal{A}_{score} = S_P + \alpha S_N + \beta S_{text}, \tag{7}$$

where $\alpha$, $\beta$ are the hyper-parameters.

## 4 Experiments

### 4.1 Experiment Setup

**Dataset and Evaluation Metrics.** We validate the effectiveness of our method across 11 datasets spanning three distinct domains: industrial, medical, and semantic domains. In the industrial domain, we utilize multiple datasets including MVTec-AD [2], Visa [46], KSDD [34], AFID [32], and ELPV [7]. For the medical domain, we incorporate datasets covering various modalities such as magnetic resonance imaging (MRI), computed tomography (CT), and optical coherence tomography (OCT). Specifically, the medical datasets include OCT2017 [17], BrainMRI, HeadCT [13], and RESC [13]. For semantic anomaly detection, we employ two datasets: MNIST [19] and CIFAR-10 [18]. These semantic datasets are utilized following the one-vs-all strategy, where one class is designated as normal, and all other classes are treated as anomalous. We follow [45] for the dataset partitions. We only use data from the MVTec-AD dataset as auxiliary training data. When testing on the MVTec-AD dataset, we use auxiliary data from the Visa dataset. The anomaly detection performance is evaluated using the Area Under the Receiver Operating Characteristic Curve (AUROC) and the Area Under the Precision-Recall Curve (AUPRC).

**Competing Methods and Baselines.** We compare our proposed method with various state-of-the-art anomaly detection (AD) methods, including traditional AD approaches (full-shot) such as PaDiM [6] and PatchCore [30], both of which are adapted to the few-normal-shot setting. Additionally, we compare a traditional few-shot learning method, RegAD [14] and prompt learning method CoOp [42], along with the latest AD methods based on vision-language models, including WinCLIP [16] and InCTRL [45]. To further validate the flexibility and applicability of our method, we evaluate our generated anomaly-free samples using four state-of-the-art anomaly detection methods: PaDiM, PatchCore, WinCLIP and InCTRL.

**Implementation Details.** We select Stable Diffusion V1.5 [29] and utilize the CLIP with ViT-L/14 architecture [28] for all tasks. Our anomaly-free customized model is fine-tuned using Dreambooth [31], and all model parameters are frozen in subsequent tasks. All the experiments are trained by use of PyTorch on an NVIDIA GeForce RTX 4090 GPU. Our anomaly detection task is a few-normal-shot setting including 2-shot, 4-shot, and 8-shot scenarios using only normal images. The ratio of the t-step is set to 0.3. We set the parameters $\alpha$ and $\beta$ for $\mathcal{A}_{score}$ to 1 and 0.5, respectively, across all datasets. The image-level memory bank is set to 30, and the inference resolution is 240×240. During inference, we use the same text prompts as WinCLIP [16]. To validate the improvement of other anomaly detection methods with our generated data, we use our generated anomaly-free samples to augment the datasets for these methods. For the MVTec dataset, we generate 100 anomaly-free samples per subclass to expand the reference dataset, while keeping all other settings identical to previous works.

### 4.2 Results

**Comparative Analysis with Industrial Datasets.** In Table 1, we comprehensively compare five different datasets from the industrial domain, with the MVTec-AD and Visa datasets containing 15 and 12 subclasses, respectively. We also conduct an in-depth analysis across various few-shot scenarios (i.e., 2-shot, 4-shot, 8-shot) to verify the robustness of our method. The experimental results demonstrate that our method outperforms other approaches, showing superior performance, especially in the 4-shot and 8-shot scenarios, where it achieves the best results across all datasets. In the 8-shot setting, our method achieves higher AUCs by 3.4%, 4.1%, 0.6%, 1.2%, and 0.9% on the KSDD, ELPV, AFID, KSDD, Visa, and MVTec datasets, respectively, compared to the second-best baseline. Notably, when compared with the latest InCTRL method on the AFID dataset, our method achieves higher AUCs by 2.2%, 3.6%, and 4.1% in the 2-shot, 4-shot, and 8-shot settings, respectively. Additionally, the results indicate that our method is less affected by different runs, with variance significantly lower than other methods, supporting its superior stability and robustness. Finally, as shown in Figure 2, the anomaly map indicates that our method accurately identifies abnormal regions

Table 1: A quantitative comparison of our proposed method and other methods using (AUROC (%), AUPRC (%)) as the evaluation metric on **industrial datasets**. We show the average performance and standard deviation across five runs, with the top value highlighted in bold for each comparison.

| | Datasets | PaDiM [6] | Patchcore [30] | RegAD [14] | CoOp [42] | WinCLIP [16] | InCTRL [45] | Ours |
|---|---|---|---|---|---|---|---|---|
| | | | | | Methods | | | |
| 2-shot | MVTec | (78.5±2.5 , 89.0±1.5) | (85.8±3.4, 93.9±1.2) | (64.0±4.7, 83.7±3.4) | (85.8±1.6, 92.2±0.7) | (93.1±1.9, 96.5±0.7) | (94.0±1.5, 96.9±0.4) | **(95.1±0.9, 97.3±0.4)** |
| | VisA | (68.0±4.2, 71.9±2.7) | (81.7±2.8, 84.1±2.3) | (55.7±5.3, 61.4±3.7) | (80.6±2.3, 83.5±1.9) | (84.2±2.4, 85.9±2.1) | (85.8±2.2, 87.7±1.6) | **(87.2±1.4, 89.1±1.3)** |
| | KSDD | (72.1±1.5, 33.7±0.8) | (90.2±0.6, 67.6±0.3) | (49.9±0.8, 17.3±1.9) | (89.7±0.6, 54.3±0.4) | (94.2±0.6, 86.5±0.4) | **(97.2±2.9, 91.7±0.9)** | (96.8±1.5, 91.6±0.6) |
| | AFID | (78.4±2.8, 52.9±3.4) | (73.9±1.7, 37.8±0.8) | (56.4±7.2, 27.5±3.5) | (68.7±6.2, 44.3±0.5) | (72.6±5.5, 50.0±4.3) | (76.1±2.9, 51.9±2.2) | **(78.3±1.7, 53.6±1.2** |
| | ELPV | (59.4±8.3, 70.7±5.8) | (71.6±3.1, 84.0±3.1) | (57.1±1.6, 67.9±0.5) | (76.2±1.1, 84.1±0.2) | (72.6±2.0 , 84.9±1.0) | (83.9±0.3, 91.3±0.8) | **(85.6±0.5, 92.0±0.5)** |
| 4-shot | MVTec | (80.5±1.8, 90.9±1.3) | (88.5±2.6, 95.0±1.3) | (66.3±3.2, 84.6±2.6) | (87.4±1.7, 92.4±0.8) | (94.0±2.1, 96.8±0.8) | (94.5±1.8, 97.2±0.6) | **(95.6±1.2, 97.8±0.6)** |
| | VisA | (73.5±3.1, 75.8±1.8) | (84.3±2.5, 86.0±1.6) | (57.4±4.2, 62.8±3.4) | (81.8±1.8, 84.2±1.6) | (85.8±2.5, 87.5±2.3) | (87.7±1.9, 90.2±2.7) | **(88.6±1.6, 90.7±0.6)** |
| | KSDD | (74.2±1.4, 35.1±1.2) | (92.3±0.8, 70.3±1.3) | (52.5±2.7, 17.6±0.3) | (90.2±0.6, 59.4±1.4) | (94.3±0.4, 86.8±0.3) | (97.5±0.6, 92.4±1.5) | **(97.8±0.8, 92.4±0.7)** |
| | AFID | (78.7±3.8, 54.0±5.3) | (73.3±0.2, 37.7±0.1) | (59.6±7.4, 29.4±3.1) | (72.0±1.7, 45.4±1.4) | (76.4±2.5, 51.3±1.7) | (79.0±1.8, 54.8±1.6) | **(82.6±0.7, 57.9±1.3)** |
| | ELPV | (61.2±0.8, 72.4±6.7) | (75.6±7.3, 87.1±4.2) | (59.6±4.0, 68.8±1.8) | (78.1±0.2, 86.7±0.3) | (75.4±0.9, 86.4±0.4) | (84.6±1.1, 91.6±0.9) | **(87.3±0.8, 92.8±0.6)** |
| 8-shot | MVTec | (82.0±1.6, 92.7±1.2) | (92.2±1.9, 96.2±1.3) | (67.4±3.3, 85.5±2.1) | (88.0±1.4, 93.3±0.7) | (94.7±2.5, 97.3±0.9) | (95.3±1.3, 97.7±0.6) | **(96.2±0.8, 98.9±0.8 )** |
| | VisA | (76.8±3.2, 78.1±2.4) | (86.0±2.6, 87.3±2.2) | (58.9±4.0, 64.3±3.2) | (82.2±2.1, 84.8±2.0) | (86.8±2.0, 88.0±2.1) | (88.7±2.1, 90.4±2.5) | **(89.9±1.3, 91.0±0.8)** |
| | KSDD | (76.9±3.7, 38.4±4.5) | (92.5±0.3, 70.8±0.9) | (59.4±2.9, 24.6±3.1) | (89.9±0.5, 57.8±0.1) | (94.1±0.1, 86.5±0.1) | (97.8±0.6, 92.5±1.1) | **(98.4±0.7, 92.7±0.8)** |
| | AFID | (79.2±2.5, 55.5±3.1) | (74.5±0.2, 38.9±0.3) | (60.3±6.2, 31.4±3.6) | (76.9±0.8, 51.4±0.3) | (79.6±1.5, 56.2±2.1) | (80.6±3.6, 56.1±3.4) | **(84.7±0.9, 63.3±1.5)** |
| | ELPV | (72.4±1.7, 79.8±1.4) | (83.7±1.6, 91.5±0.7) | (63.3±2.7, 69.6±1.5) | (81.7±1.2, 90.5±0.8) | (81.4±1.0, 89.7±0.7) | (87.2±1.3, 92.6±0.6) | **(90.6±0.9, 94.1±0.8)** |

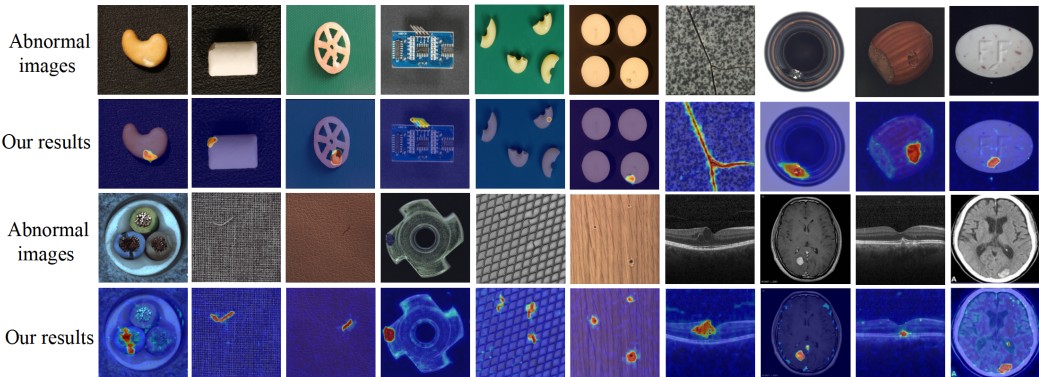

Figure 2: Visualization of representative results for pixel-level anomaly localization of our proposed method on different datasets.

while being less likely to falsely recognize normal regions as anomalous. This further validates the accuracy of our approach. More anomaly maps and localization results are provided in Appendix A.2.

**Comparative Analysis with Medical Datasets.** In Table 2, we compare four medical image datasets from three different modalities (CT, MRI, OCT). Our method achieves superior few-shot anomaly detection performance across these diverse modalities, outperforming other methods in most datasets, particularly on the OCT2017 and RESC datasets. Notably, on the RESC dataset, our method surpasses the second-best method by 3.9%, 5.9%, and 4.6% in the 2-shot, 4-shot, and 8-shot settings, respectively. The suboptimal performance of other baseline methods on these datasets may be due to the unique characteristics of the medical domain, where much of the knowledge is not well-explored in VLMs pre-trained mostly on non-medical data. These results demonstrate that our proposed anomaly personalization strategy alleviates these domain-specific challenges more effectively. Additionally, because our method is highly effective in specific domains, it demonstrates greater stability compared to other methods, further proving its robustness.

**Comparative Analysis with Semantic Datasets.** Here, we compare two semantic datasets using the one-vs-rest strategy, where one class is considered normal, and all other classes are treated as anomalous. The results show that our method achieves the best performance across three different few-shot settings. In the 8-shot scenario, our method reaches AUROC scores of 93.6(%) and 94.9(%) on the MNIST and CIFAR-10 datasets, respectively. These experiments demonstrate that our method is also more effective on semantic datasets compared to other baselines.

Table 2: A quantitative comparison of our proposed method and other methods using (AUROC (%), AUPRC (%)) as the evaluation metric on **medical datasets**. We show the average performance and standard deviation across five runs, with the top value highlighted in bold for each comparison.

| | | | | Methods | | | |
|---|---|---|---|---|---|---|---|
| | Datasets | PaDiM [6] | Patchcore [30] | RegAD [14] | WinCLIP [16] | InCTRL [45] | Ours |
| 2-shot | OCT2017 | - | (73.0±3.6, 86.6±2.1) | (70.1±5.2, 84.6±3.9) | (94.7±2.2, 98.3±0.7) | (94.9±2.6, 98.3±2.4) | (**96.7±0.9, 99.0±1.4**) |
| | BrainMRI | (65.7±12.2, 90.2±4.6) | (70.6±0.9, 92.1±1.7) | (44.9±12.9, 87.2±6.5) | (93.4±1.2, 98.9±0.3) | (**97.3±2.7, 99.4±1.3**) | (97.1±1.4, 99.3±0.9) |
| | HeadCT | (59.5±3.6, 87.6±1.7) | (73.6±9.6, 91.3±0.2) | (60.2±1.8, 85.4±0.9) | (91.5±1.5, 97.5±1.2) | (92.9±2.5, 98.1±1.3) | (**94.2±1.1, 98.6±1.3**) |
| | RESC | - | (69.3±5.4, 66.2±3.4) | (69.2±3.9, 65.8±3.6) | (85.46±2.1, 79.5±0.4) | (88.3±3.4, 81.48±2.5) | (**92.2±0.8, 84.3±1.2**) |
| 4-shot | OCT2017 | - | (76.8±2.8, 88.1±2.0) | (72.68±3.1, 86.14±3.8) | (96.2±2.7, 98.6±0.5) | (96.8±2.4, 98.9±2.4) | (**99.1±1.2, 99.5±1.4** ) |
| | BrainMRI | (79.2±4.8, 95.6±1.1) | (79.4±4.0, 94.5±1.7) | (57.1±14.9, 90.0±4.1) | (94.1±0.2, 99.0±0.1) | (97.5±1.6, 99.4±1.3) | (**98.2±1.3, 99.6±0.5**) |
| | HeadCT | (62.2±1.3, 89.0±1.1) | (80.5±0.6, 94.1±0.9) | (52.2±5.0, 81.0±2.8) | (91.2±0.3, 97.4±0.2) | (93.3±1.3, 98.4±1.1) | (**94.7±0.6, 99.0±0.7**) |
| | RESC | - | (69.5±3.4, 66.8±2.1) | (68.5±2.7, 65.3±2.5) | (87.9±2.1, 80.8±1.3) | (88.7±2.3, 81.1±2.4) | (**94.6±1.3, 93.2±0.9**) |
| 8-shot | OCT2017 | - | (80.6±2.1, 90.4±1.6) | (74.38±2.9, 88.6±4.7) | (97.0±2.9, 99.0±0.5) | (97.4±2.1, 99.1±1.7 ) | (**99.3±1.5, 99.7±0.8**) |
| | BrainMRI | (75.8±2.5, 94.6±0.7) | (81.2±1.6, 95.7±0.7) | (63.2±7.9, 90.8±1.3) | (94.4±0.1, 99.1±0.0) | (98.3±1.2, 99.6±0.3) | (**98.6±0.7, 99.6±0.5**) |
| | HeadCT | (66.1±3.9, 89.6±0.9) | (81.7±3.4, 93.1±0.6) | (62.8±2.6, 88.1±1.4) | (91.5±0.8, 97.5±0.3) | (93.6±0.8, 98.5±0.5) | (**94.8±0.6, 99.1±0.6**) |
| | RESC | - | (71.2±2.4, 67.9±2.9 ) | (68.7±3.2, 65.5±1.9) | (88.92±2.9, 83.1±1.6) | (90.6±2.7, 83.4±1.8) | (**95.2±1.4, 87.6±1.8**) |

Table 3: A quantitative comparison of our proposed method and other methods using (AUROC (%), AUPRC (%)) as the evaluation metric on **semantic datasets**. We show the average performance and standard deviation across five runs, with the top value highlighted in bold for each comparison.

| | | | | Methods | | | |
|---|---|---|---|---|---|---|---|
| | Datasets | Patchcore [30] | RegAD [14] | CoOp [42] | WinCLIP [16] | InCTRL [45] | **Ours** |
| 2-shot | MNIST | (75.6±0.4, 95.6±0.1) | (52.5±3.0, 91.3±0.6) | (55.7±0.6, 92.6±0.3) | (81.0±0.8, 96.3±0.1) | (89.2±0.9, 97.5±0.4) | (**92.6±0.5, 98.9±0.4**) |
| | CIFAR-10 | (60.2±0.9, 92.6±0.2) | (53.4±0.5, 90.9±0.3) | (52.7±1.1, 91.1±0.2) | (92.5±0.1, 99.0±0.1) | (93.5±0.2, 99.2±0.0) | (**94.1±0.2, 99.3±0.1**) |
| 4-shot | MNIST | (83.3±0.9, 97.2±0.2) | (54.8±5.3, 91.6±1.3) | (56.3±0.4, 92.9±0.2) | (85.1±1.0, 97.1±0.2) | (90.2±1.6, 98.0±0.7) | (**92.9±0.7, 99.0±0.6**) |
| | CIFAR-10 | (63.9±1.0, 93.4±0.3) | (53.4±0.2, 90.8±0.1) | (53.7±0.5, 91.5±0.3) | (92.7±0.1, 99.0±0.0) | (94.0±1.0, 99.2±0.4) | (**94.6±0.2, 99.3±0.2**) |
| 8-shot | MNIST | (87.6±0.4, 97.9±0.1) | (54.7±6.3, 91.9±1.8) | (56.7±0.7, 93.7±0.4) | (86.7±0.7, 97.4±0.1) | (92.0±0.3, 98.9±0.1) | (**93.6±0.2, 99.5±0.3**) |
| | CIFAR-10 | (67.2±0.6, 94.2±0.2) | (55.5±0.8, 91.1±0.1) | (54.2±0.5, 92.0±0.3) | (92.8±0.1, 99.0±0.0) | (94.5±0.2, 99.4±0.1) | (**94.9±0.1, 99.5±0.1**) |

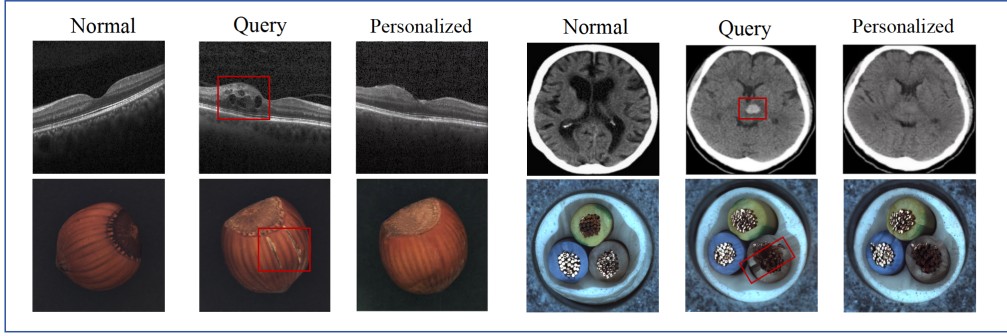

Figure 3: Visualizations of anomaly personalization. The red box of the query image indicates anomalous regions.

**Visualizations of Anomaly Personalization.** Figure 3 compares the normal images (i.e., reference images), query images, and our resulting personalized images. As depicted, there are some differences between the normal image and the query image in terms of non-anomalous features, such as position, shape, and texture. Our personalized image retains most of the normal regions from the query image, with the anomalous regions largely converted to normal regions. This visualization further substantiates the precision of our method compared to existing few-shot AD approaches.

## 4.3 Ablation Study

**The effectiveness of generated anomaly-free samples.** To validate the effectiveness of our method in generating normal samples, we apply it to multiple AD methods, including Patchcore, RegAD, WinCLIP and InCTRL. As shown in the first row of Figure 4, our method demonstrates improvements

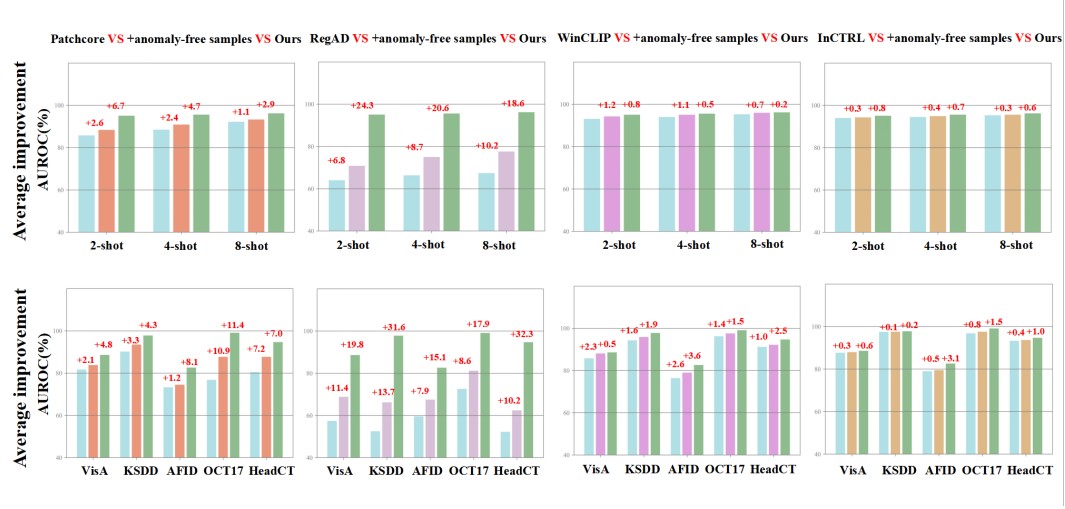

Figure 4: The effectiveness of our generated anomaly-free samples for other AD methods. The red numbers highlight performance increases.

in the 2-shot, 4-shot, and 8-shot settings across all three methods. Notably, for the RegAD method, the AUC is improved by 6.8(%), 8.7(%), and 10.2(%), respectively. Additionally, we compare our generated strategy with five other AD datasets. After incorporating our anomaly-free samples, the four AD methods show varying degrees of improvement on industrial and medical datasets (Visa, KSDD, AFID, OCT2017, and headCT). In particular, the Patchcore and RegAD methods exhibit significant improvements on the OCT2017 dataset, with increases of 10.9(%) and 8.6(%), respectively. While InCTRL has already achieved high performance, our method still provides a certain level of enhancement. This experiment demonstrates that the data generated by our method can enhance existing AD methods, proving the flexibility and adaptability of our approach.

**The effectiveness of the triplet contrastive anomaly inference.** Here, we discuss the impact of each branch in our triplet contrastive anomaly inference strategy, analyzing datasets across three different domains. Table 4 validates the effectiveness of our strategy, showing that it is optimal in most datasets. When only text prompts are used ($S_{text}$), meaning there is no reference image input, particularly low results are observed in the medical datasets OCT2017 and RESC. However, when reference images are supplied (either $S_P$ or $S_N$), a significant improvement is observed. This might be because the model lacks an inherent understanding of specific medical domains, thus requiring normal images as references. When each strategy is used individually, we find that most results are not as good as when the strategies are combined, indicating that these strategies complement each other. When all three strategies are employed together, most datasets demonstrate optimal performance. However, in certain cases, such as the KSDD dataset—which focuses on surface defect inspection—utilizing all scores does not consistently yield the best results. This inconsistency may stem from the high diversity in the distribution of normal images, which presents challenges for the AD model's learning process. Nevertheless, across the majority of datasets, the combined use of three strategies leads to relatively strong performance, showcasing that our triplet contrastive anomaly inference strategy effectively enhances accuracy.

**The effectiveness of text prompts in anomaly personalization.** We also conduct ablation experiments on the settings of the text prompts $c_q$ used in the one-to-normal personalization stage (Section 3.3). In this work, we employed three prompts to generate three images (one for each prompt). The results in Table 5 demonstrate that our designed text prompt strategy is effective for one-to-normal personalization, where most datasets show improvement.

**Inference time.** Regarding inference time, our proposed method is slightly higher (+200-300ms per query image) than that of WinCLIP (389ms) and InCTRL (276ms). If necessary, we can further increase the inference speed by reducing the number of generated samples or decreasing the memory bank size. When using a single prompt corresponds to generating only one personalized image, the

Table 4: Ablation study on the impact of three different contrastive anomaly inference branches in 8-shot setting. We show the average performance (AUROC (%)) across five runs, with the top value highlighted in bold.

| Strategies | | | Datasets | | | | | | | | | | |
|---|---|---|---|---|---|---|---|---|---|---|---|---|---|
| $S_P$ | $S_N$ | $S_{text}$ | Industrial field | | | | | Medical field | | | | Semantic fields | |
| | | | MVTec-AD | VisA | KSDD | AFID | ELPV | OCT2017 | BrainMRI | HeadCT | RESC | MINIST | CIFAR-10 |
| | | ✓ | 91.8 | 78.2 | 94.3 | 72.8 | 73.1 | 45.3 | 92.4 | 89.6 | 39.6 | 69.2 | 91.3 |
| | ✓ | | 93.1 | 86.3 | 93.1 | 80.5 | 83.3 | 87.1 | 89.6 | 86.0 | 89.6 | 91.6 | 93.6 |
| ✓ | | | 93.7 | 88.6 | 96.3 | **85.2** | 86.1 | 96.3 | 94.3 | 90.3 | 92.6 | **93.9** | 84.3 |
| | ✓ | ✓ | 95.8 | 88.6 | 95.3 | 80.3 | 87.1 | 97.6 | **98.7** | 92.2 | 89.2 | 90.6 | 93.8 |
| ✓ | | ✓ | 96.0 | 89.6 | **98.6** | 84.3 | 88.3 | 99.2 | 94.6 | 92.6 | 94.2 | 92.6 | 88.2 |
| ✓ | ✓ | ✓ | **96.2** | **89.9** | 98.4 | 84.7 | **90.6** | **99.3** | 98.6 | **94.8** | **95.2** | 93.6 | **94.9** |

Table 5: Ablation study on the text prompts used in one-to-normal personalization on all datasets in 8-shot setting. We show the average performance (AUROC (%)) across five runs, with the top value highlighted in bold.

| | Datasets | | | | | | | | | | |
|---|---|---|---|---|---|---|---|---|---|---|---|
| Strategy | Industrial domain | | | | | Medical domain | | | | Semantic domain | |
| | MVTec-AD | VisA | KSDD | AFID | ELPV | OCT2017 | BrainMRI | HeadCT | RESC | MINIST | CIFAR-10 |
| w/o text | 95.6 | 88.6 | 98.0 | 84.6 | 89.8 | 98.7 | 97.6 | 93.9 | 94.6 | 93.5 | 93.7 |
| refined text | **96.2** | **89.9** | **98.4** | **84.7** | **90.6** | **99.3** | **98.6** | **94.8** | **95.2** | **93.6** | **94.9** |

required inference time (326ms) is slightly lower than that of WinCLIP, while still demonstrating superior performance compared to other methods.

# 5 Conclusion

We introduce a novel personalized few-shot anomaly detection method that enhances prediction accuracy through one-to-normal personalization of query images. Unlike other state-of-the-art approaches that directly compare query images with reference images, our method enables a finer-grained comparison. Our triplet contrastive anomaly inference strategy further stabilizes results by incorporating personalized images, anomaly-free samples and text prompts, facilitating a more comprehensive comparison. Extensive experiments across 11 datasets in industrial, medical, and semantic domains demonstrate the method's generalizability and effectiveness. Moreover, the anomaly-free samples generated by our method can augment the normal samples in existing few-shot anomaly detection techniques. Experimental results also demonstrate that our method improves the performance of several existing few-shot anomaly detection techniques.

**Limitation** Achieving precise detection necessitates a comprehensive exploration of the distribution of each category, which in turn requires a few reference images from each category. Consequently, our current method is not applicable to zero-shot scenarios. Future research will focus on enhancing the capability to identify anomalies in zero-shot settings more accurately, as well as exploring real-time applications and open-vocabulary scenarios.

**Broader Impacts** This article focuses on a customized few-shot anomaly detection method, which offers more precise and stable anomaly detection. It has the potential to enhance development in industrial and medical fields. There are no negative societal impacts involved in this work.

# Acknowledgement

This work was supported in part by the National Key Research and Development Program of China under Grant 2020YFB1711500 and Grant 2020YFB1711503; in part by the 1.3.5 project for disciplines of excellence, West China Hospital, Sichuan University, under Grant ZYYC21004 and ZYAI24053; in part by the Aier Eye Hospital-Sichuan University Research Grant 23JZH043.

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

# A    Appendix / supplemental material

## A.1    Text Prompt Setting

The text prompts are provided in Figure 5

(a) *State*-level (-:normal)

- c := "[o] without flaw"
- c := "[o] without defect"
- c := "[o] without damage"

(b) *Physical*-level

- "a photo of a/the small [c]."
- "a photo of a/the large [c]."
- "a bright photo of a/the [c]."
- "a dark photo of a/the [c]."
- "a blurry photo of a/the [c]."

- "a bad photo of a/the [c]."
- "a good photo of a/the [c]."
- "a cropped photo of a/the [c]."
- "a close-up photo of a/the [c]."
- "a low resolution photo of a/the [c]."

Figure 5: Lists of state and template level prompts employed in this paper to construct text features.

Our experimental approach initially focused on the number of prompts, investigating scenarios with 1, 3, 5, and 10 prompts. We found that performance was at its lowest with a single prompt, whereas using all ten prompts resulted in the highest performance.

## A.2    Abnormal Localization Results

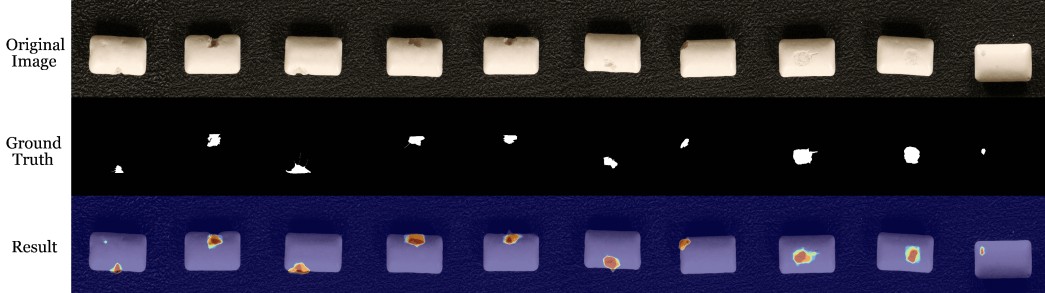

Figure 6: The anomaly map localization results of our proposed method for subclass chewinggum. The first row shows the original images with anomalies, the second row displays the ground truth of the anomalies, and the third row shows the localization results obtained by our method.

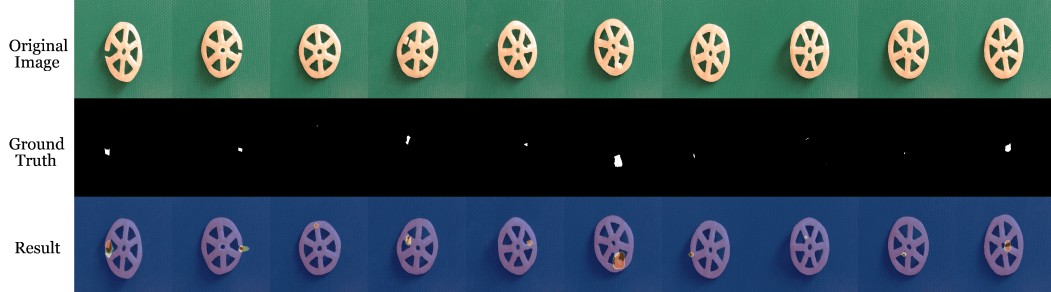

Figure 7: The anomaly map localization results of our proposed method for subclass fryum. The first row shows the original images with anomalies, the second row displays the ground truth of the anomalies, and the third row shows the localization results obtained by our method.

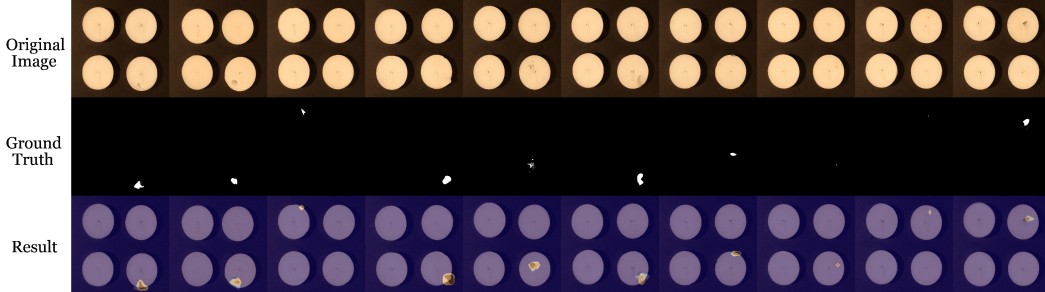

Figure 8: The anomaly map localization results of our proposed method for subclass candle. The first row shows the original images with anomalies, the second row displays the ground truth of the anomalies, and the third row shows the localization results obtained by our method.

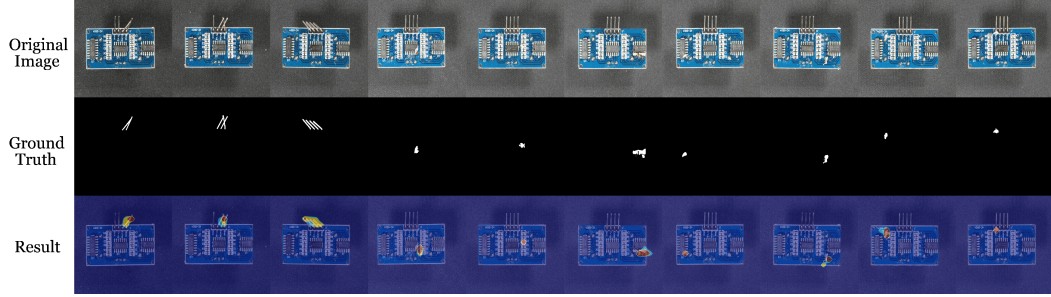

Figure 9: The anomaly map localization results of our proposed method for subclass pcb2. The first row shows the original images with anomalies, the second row displays the ground truth of the anomalies, and the third row shows the localization results obtained by our method.

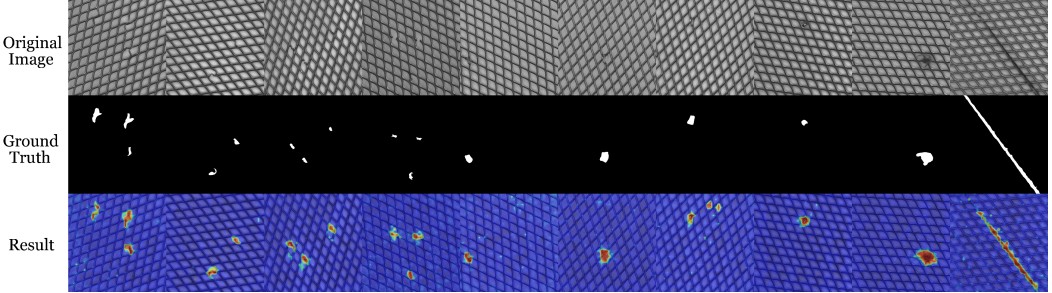

Figure 10: The anomaly map localization results of our proposed method for subclass grid. The first row shows the original images with anomalies, the second row displays the ground truth of the anomalies, and the third row shows the localization results obtained by our method.

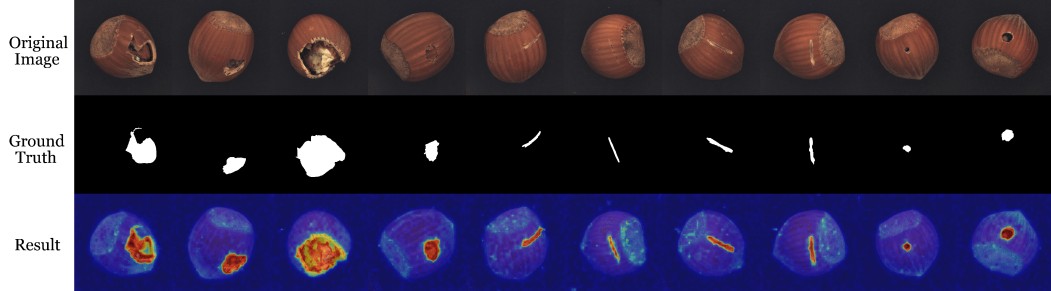

Figure 11: The anomaly map localization results of our proposed method for subclass hazelnut. The first row shows the original images with anomalies, the second row displays the ground truth of the anomalies, and the third row shows the localization results obtained by our method.

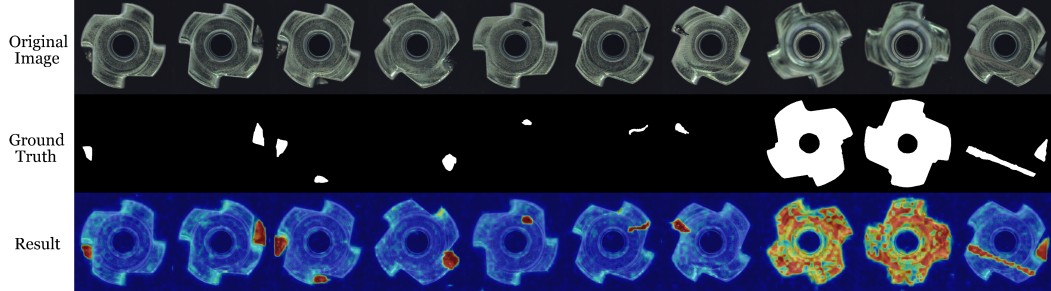

Figure 12: The anomaly map localization results of our proposed method for subclass metal_nut. The first row shows the original images with anomalies, the second row displays the ground truth of the anomalies, and the third row shows the localization results obtained by our method.

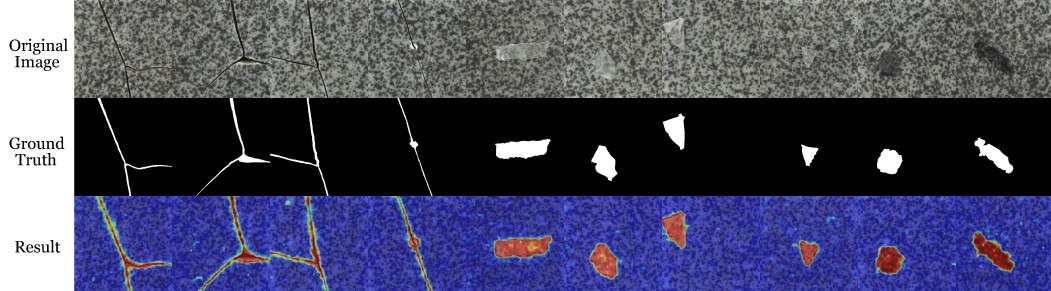

Figure 13: The anomaly map localization results of our proposed method for subclass tile. The first row shows the original images with anomalies, the second row displays the ground truth of the anomalies, and the third row shows the localization results obtained by our method.

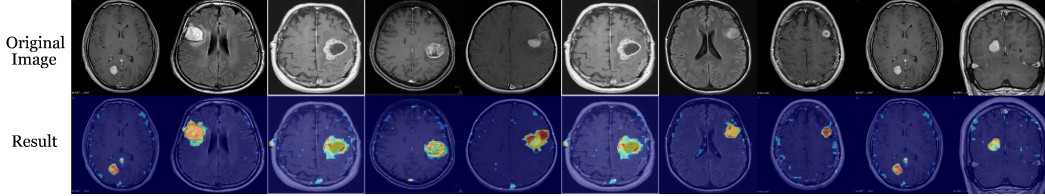

Figure 14: The anomaly map localization results of our proposed method for subclass BrainCT. The first row shows the original images with anomalies, the second row displays the ground truth of the anomalies, and the third row shows the localization results obtained by our method.

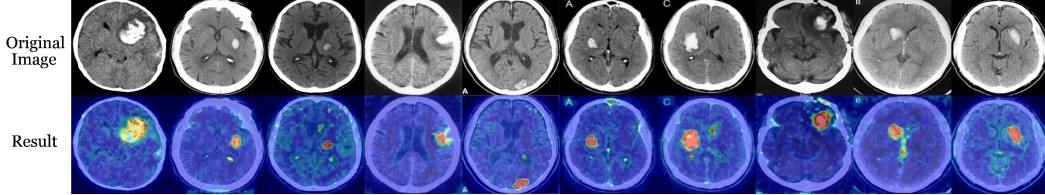

Figure 15: The anomaly map localization results of our proposed method for subclass HeadCT. The first row shows the original images with anomalies, the second row displays the ground truth of the anomalies, and the third row shows the localization results obtained by our method.

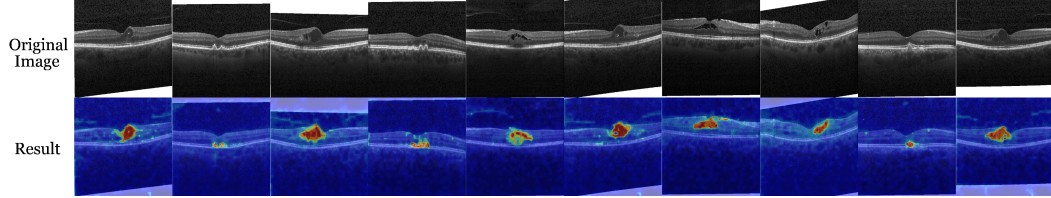

Figure 16: The anomaly map localization results of our proposed method for subclass OCT. The first row shows the original images with anomalies, the second row displays the ground truth of the anomalies, and the third row shows the localization results obtained by our method.

