# OpenReview forum: "One-to-Normal: Anomaly Personalization for Few-shot Anomaly Detection"
_NeurIPS.cc/2024/Conference — NeurIPS 2024 poster_

### Official Review · Reviewer_WVvk · 2024-07-01

**Soundness:** 3
**Presentation:** 3
**Contribution:** 3
**Rating:** 7
**Confidence:** 3

**Summary:**

This paper provides a novel method to tackle the issue of precision loss in more complex domains. They introduce an anomaly personalization method with a diffusion model to utilize the diffusions to obtain the normal sample distribution and exchange the anomaly image into normal ones. Finally, a triplet contrastive strategy is designed to obtain the score. Their approach obtains a SOTA result compared to the recent methods.

**Strengths:**

1. This paper provides novel approaches that utilize the diffusion model for both distribution modeling and recovering the anomaly images into the real one for comparison.
2. The triplet contrastive strategy introduces a multi-level way to obtain the anomaly map from different views which is more robust than the previous methods.
3. The experiments are comprehensive and the result is SOTA compared to the recent methods.

**Weaknesses:**

1. Will the utilize of the diffusion model bring a long inference time, which may not be effective for real-time application?
2. Will the choice of the text influence the result? Is there any comparison result with different prompt choices?

**Questions:**

See Weakness.

**Limitations:**

The authors have address the limitations in zero-shot scenarios.

---

> ### Author Rebuttal · Authors · 2024-08-07
>
> We thank the reviewer for the careful reviews and constructive suggestions. We answer the questions as follows.
> ***
>
> >W#1: Will the utilize of the diffusion model bring a long inference time, which may not be effective for real-time application
>
> **A#1:** Thank you for the constructive question. 1) Our approach has a relatively short diffusion process and inference time compared to most diffusion-based anomaly detection methods, because our diffusion step ratio is only 0.3. 2) For very recent anomaly detection methods that do not utilize diffusion models (e.g., WinCLIP and InCTRL), the inference time of our proposed method is slightly higher (+200-300ms per query image) than that of WinCLIP (389ms) and InCTRL (276ms). 3) If necessary, we can further decrease the inference time by reducing the number of generated samples or decreasing the memory bank size. When using a single prompt corresponds to generating only one personalized image, the required inference time (326ms) is slightly lower than that of WinCLIP, while still demonstrating superior performance across three domains compared to other methods.
>
> To improve the application in real-time scenarios, the following strategies may be considered: 1) Performing feature-level comparisons to reduce the steps involved in encoding and generating images. 2) Exploring the possibility of model pruning to lower computational complexity. 3) Employing more efficient diffusion model architectures (e.g., AT-EDM). Further exploration can be pursued in our future work.
>
>
>
> ***
>
> ***
>
> >W#2: Will the choice of the text influence the result? Is there any comparison result with different prompt choices?
>
> **A#2:** Thank you for your insightful question. Yes, the choice of text will influence the results. In our preliminary research, we conducted a comparison experiment. Our experiments explored both the category and quantity of prompts:
>
> 1) Exploration by Category: We categorized text prompts based on physical-level attributes related to the image, such as global quality (e.g., "a good photo of a/the [c]"), object size (e.g., "a photo of a/the small [c]"), and image resolution (e.g., "a low resolution photo of a/the [c]"). We tested the performance using prompts from a single category, two categories, and all three categories. We found that using prompts from all three categories resulted in the best performance. For your convenience, we have provided some experimental results on the MVTec-AD dataset with three prompts as a reference below.
>
> | Datasets   | Global quality | Object size | Image resolution | AUROC |
> |------------|----------------|-------------|------------------|-------|
> |            |                | ✅           |                  | 95.8  |
> | **MVTec-AD** |                |             | ✅                | 95.9  |
> |            | ✅              |             |                  | 95.9  |
> |            |                | ✅           | ✅                | 96.0  |
> |            | ✅              | ✅           | ✅                | 96.2  |
>
> 2）Exploration by quantity: We examined scenarios with 1, 3, 5, and 10 prompts. We observed that when there was only one prompt, the performance was the lowest, whereas using all ten prompts yielded the highest performance. Generally, we find there is no significant difference in performance between using three and five prompts.
>
>
> | Numbers of prompt | AUROC |
> |:-----------------:|:-----:|
> |         1         | 95.9  |
> |         3         | 96.2  |
> |         5         | 96.2  |
> |        10         | 96.4  |
>
> Our results suggest that the more comprehensive the inclusion of these three categories, the better the performance, with the optimal scenario using all 10 prompts. However, considering efficiency, we aimed to achieve the best results with the fewest prompts possible. We selected one prompt from each category and discovered through experiments that the combination of "a good photo of a/the [c]," "a close-up photo of a/the [c]," and "a low resolution photo of a/the [c]" yielded the best results in most cases. This combination of three prompts is what we present in the main figure describing our method in the manuscript.
>
>
>
>
> ***

---

> > ### Comment · Reviewer_WVvk · 2024-08-12
> > **Thank you for your feedback!**
> >
> > Thank you for your feedback. I think your reply solve my concerns. I will keep my rate as Accept.

---

> > > ### Author Response · Authors · 2024-08-12
> > >
> > > We sincerely appreciate you taking the time to review our rebuttal and helpful feedback on our work. We are very glad that our response has addressed your concerns.

---

### Official Review · Reviewer_oVNv · 2024-07-07

**Soundness:** 3
**Presentation:** 3
**Contribution:** 3
**Rating:** 5
**Confidence:** 3

**Summary:**

This paper proposes a new few-shot anomaly detection method based on one-to-normal personalization of query images using a diffusion model and a triplet contrastive inference process. It leverages a diffusion-based generation model to transform a query image into a personalized image towards the distribution of normal images, for use in the triplet contrastive inference. Experimental results on datasets from industrial, medical, and semantic domains demonstrate that the proposed method outperforms existing models.

**Strengths:**

- Unlike common augmentation-based methods that generate pseudo-anomalies for AD, this approach transforms a query image towards the normal distribution, which is interesting and effective. The proposed triplet contrastive inference is compelling.

- Experiments conducted across various domains show its state-of-the-art performance in few-shot scenarios.

**Weaknesses:**

- Details about the experimental settings are not fully provided, which limits reproducibility. For instance, there is no information on the selection of hyperparameters such as alpha and beta for A_score in the implementation details. It raises concerns about the model's performance. If alpha and beta were optimized based on test data, the claim of state-of-the-art performance is questionable. Details about the memory bank M such as its size and sensitivity of the performance to its hyperparameters are also missing.
- The paper lacks theoretical analysis, relying solely on empirical evidence.
- Figure 1 does not provide sufficient information about the overall process. Particularly, it omits the training process of the diffusion model and the composition of the anomaly-free pool, which can cause confusion before reading the detailed explanations in the main text.
- The definition of S_text in Sec 3.4 is unclear. It does not seem logical to apply the softmax function directly to the paired feature of F_q,l and F^T_text.
- The lack of equation numbers in the Method section makes it difficult to follow.
- There is no discussion of Table 3 in the paper.

**Questions:**

- Can you provide visualizations or information about the distribution of the individual anomaly scores (S_n, S_p, S_text)?  It would be helpful to understand whether these scores focus on different parts or the same regions.
- The sensitivity of the model to text prompts should be further explored. Specifically, I would like to know the number of selected text prompts when generating personalized image and the rationale or justification for setting physical-level text prompts in the chosen template.
- In Table 4, using all the scores does not always correspond to the highest performance. A discussion on the possible reasons for this would be valuable.
- Could you provide results from InCTRL in Figure 4, and also add lines or bars to compare each performance against the performance of the proposed model?
- Why is the performance on MVTec-AD different in Table 5 (96.8) and Table 4 (96.2)?

Minor issues:
- In Table 4, the top value for AFID is 85.2, not 84.7 as bold-faced.
- Tables 4 and 5 lack information on the few-shot setting.
- Typo: "semantice" should be "semantic" in the caption of Table 3.

**Limitations:**

- As mentioned in the paper, the method is not applicable in zero-shot scenarios.
- The theoretical exploration is lacking; there is only empirical evidence.
- The computational cost is likely to be high, which may not be suitable for real-world scenarios
- The approach depends on the performance of the generative model. And it is limited if the generated images are abnormal (potentially vulnerable to attacks).

---

> ### Author Rebuttal · Authors · 2024-08-07
>
> ***
>
> >W#1: Details about the experimental settings are not fully provided, which limits reproducibility. The hyperparameters such as alpha and beta for $A_score$ and Details about the memory bank M.
>
> **A#1:** Thank you for your constructive comments. 1) We set the parameters $\alpha$ and $\beta\$ for $A_{\text{score}}$ to 1 and 0.5, respectively. This configuration remains consistent across all datasets and is not optimized based on the test data of each dataset. To show the robustness of our method to different choices of hyperparameters, we present the detailed results in Table 1 of **the uploaded PDF**. We will also include more details in the revision.
> 2) The memory bank size has been set to 30. Our preliminary experiments (Figure 1 in **the uploaded PDF**) indicate that larger memory bank values tend to improve results, but the improvement saturates after reaching a certain threshold. To balance model efficiency and performance, we selected \(M = 30\). For clarity, we have updated the manuscript to include the specific values of these parameters.
>
>
>
>
> ***
>
> ***
>
> >W#2: Figure 1 does not provide sufficient information about the overall process. Particularly, it omits the training process of the diffusion model and the composition of the anomaly-free pool, which can cause confusion before reading the detailed explanations in the main text.
>
>
> **A#2:** We apologize for any confusion caused. 1)The training process of the diffusion model follows the DreamBooth fine-tuning method to customize a diffusion model. We will provide more details in the revision.  2) Our anomaly-free sample pool comprises a set of normal reference images and generated normal images as mentioned in Section 3.4 of the manuscript. To avoid confusion, we will include a detailed explanation in the revision.
>
>
> ***
>
>
> ***
>
> >W#3: The definition of S_text in Sec 3.4 is unclear. It does not seem logical to apply the softmax function directly to the paired feature of F_q,l and F^T_text.
>
>
>
> **A#3:** Thank you for pointing this out and identifying the typo in our manuscript. $S_{text}$ determines the anomaly score by assessing the similarity between the text prompt and the query image. The image and text features are in fact combined using a dot product, and a softmax function calculates the probability, which serves as the anomaly score. Thank you for your correction; we have updated the manuscript accordingly.
>
>
> ***
>
> ***
>
> >Q#1: Can you provide visualizations or information about the distribution of the individual anomaly scores (S_n, S_p, S_text)? It would be helpful to understand whether these scores focus on different parts or the same regions.
>
>
> **A#4:** Thank you for the constructive questions. Following your suggestion, we have provided visualizations of the distribution of the individual anomaly scores. As illustrated in the figure of **the uploaded PDF**, combining both personalization and text prompts leads to the best performance, with each focusing on different regions.
>
>
> ***
> ***
>
> >Q#2: The sensitivity of the model to text prompts should be further explored. Specifically, I would like to know the number of selected text prompts when generating personalized image and the rationale or justification for setting physical-level text prompts in the chosen template.
>
>
> **A#5:** We thank you for your constructive suggestion. In this study, we utilized three text prompts, derived from our prior experiments. Our experimental approach initially focused on the number of prompts, investigating scenarios with 1, 3, 5, and 10 prompts. We found that performance was at its lowest with a single prompt, whereas using all ten prompts resulted in the highest performance.
> | Numbers of prompt | AUROC |
> |:-----------------:|:-----:|
> |         1         | 95.9  |
> |         3         | 96.2  |
> |         5         | 96.2  |
> |        10         | 96.4  |
>
> The rationale for employing physical-level text prompts is predicated on the assumption that these prompts possess attributes directly related to the image, categorized into three main groups: global quality (e.g., 'a good photo of a/the [c]'), object size (e.g., 'a photo of a/the small [c]'), and image resolution (e.g., 'a low resolution photo of a/the [c]'). We assessed performance using prompts from one, two, or all three categories. Our results demonstrated that the prompts from all three categories yielded the best performance, leading us to adopt three prompts encompassing these categories for our analysis.
>
>
> ***
>
>
>
>
> ***
>
> >Q#3: In Table 4, using all the scores does not always correspond to the highest performance. A discussion on the possible reasons for this would be valuable.
>
>
> **A#6:** Yes, thank you for your valuable suggestion. Using all scores generally achieves the highest performance across most datasets, but it does not always correspond to the highest performance on a few datasets, for example, the KSDD dataset, a surface defect inspection dataset.  One possible explanation might be the high diversity in the distribution of normal images for the diffusion model to learn. In contrast, the BrainMRI medical dataset consists of grayscale images and features highly symmetrical normal samples. Such symmetry is not fully manifested in the personalized images, leading to better results when only using generated images and text.
>
> ***
>
> ***
>
> >Q#4: Could you provide results from InCTRL in Figure 4, and also add lines or bars to compare each performance against the performance of the proposed model?
>
>
> **A#7:** Thank you for your helpful suggestion. In Figure 2 of **the uploaded PDF**, we present results from InCTRL and include lines or bars to compare each performance against the performance of the proposed method.
>
> ***
> ***
> >Q#5: The lack of equation number, MVTec-AD performance typo, and other minor issues:
>
>
> **A#8:** We greatly appreciate your careful review to improve our manuscript. We will address all issues in the revision.

---

### Official Review · Reviewer_tpYi · 2024-07-10

**Soundness:** 3
**Presentation:** 3
**Contribution:** 3
**Rating:** 5
**Confidence:** 3

**Summary:**

This paper addresses the issue of few-shot anomaly detection, which introduces an anomaly personalization method by using an anomaly-free customized generation model and performing a triplet contrastive anomaly inference strategy. Experiment evaluations across eleven datasets in three domains demonstrate its superior performance compared to the latest AD methods.

**Strengths:**

1. The paper is generally well-organized and well-written.

2. The ideas of anomaly personalization and triplet contrastive anomaly inference are well-motivated with solid theoretical support.

3. The superior performances on various evaluations demonstrate the effectiveness of the proposed method.

**Weaknesses:**

- There is a lack of discussion of the computation cost and inference speed of the proposed method since there are many generation steps. And the explicit description and exact number of prompts and images for generation are not clearly stated.

- The method introduces several hyperparameters (e.g., α, β) which require careful tuning. It's unclear how sensitive the results are to these hyperparameters and whether the paper provides sufficient guidance on setting them. Regarding the process of multi-level feature comparison, it would be better to clarify the effect of the number of multi-feature extraction blocks.

- The equations are not numbered and the line numbers are incomplete which makes it hard to reference.

**Questions:**

1. What are the exact components of $C_0$ and $\overline{C_0}$? As for the normal state prompts, is the number of $n$ below Line125 13? And what is the number of the memory bank $M$?

2. The "Triplet Contrastive Anomaly Inference" seems to work as a weighted prediction from three comparison aspects during testing. Where can it reflect the concept of "contrastive"? What is the full training objective?

3. Does $S_{text}$ reflect the degree of anomaly score? It seems that there are two types (both normal and abnormal) objects in the text prompts. How do they work in the same way as the equations of $S_{text}$ and $A_{score}$?

4. Is there any limitation for the proposed method to handle the open-vocabulary scenarios? And what is the computation cost and inference cost of the proposed method compared with other methods since there are many images for generation?

**Limitations:**

The authors provide a brief discussion of the limitations on zero-shot anomaly detection.

---

> ### Author Rebuttal · Authors · 2024-08-07
>
> We thank the reviewer for the careful reviews and constructive suggestions. We answer the questions as follows.
> ***
>
> >W#1: There is a lack of discussion of the computation cost and inference speed of the proposed method since there are many generation steps. And the explicit description and exact number of prompts and images for generation are not clearly stated.
>
>
> **A#1:** We thank the reviewer for the constructive question. In this work, we employed three prompts to generate three images (one for each prompt), and the inference time of our proposed method is slightly higher (+200-300ms per query image) than that of WinCLIP (389ms) and InCTRL (276ms). If necessary, we can further increase the inference speed by reducing the number of generated samples or decreasing the memory bank size. When using a single prompt corresponds to generating only one personalized image, the required inference time (326ms) is slightly lower than that of WinCLIP, while still demonstrating superior performance across three domains compared to other methods.
>
>
> ***
>
> ***
>
> >W#2: The method introduces several hyperparameters (e.g., α, β) which require careful tuning. It's unclear how sensitive the results are to these hyperparameters and whether the paper provides sufficient guidance on setting them. Regarding the process of multi-level feature comparison, it would be better to clarify the effect of the number of multi-feature extraction blocks
>
> **A#2:** Thank you for your detailed question. The hyperparameters α and β are set to 1 and 0.5, respectively, across all experiments and datasets. This setting was determined based on our preliminary experiments. For your convenience, we have presented these results in Table 1 of **the uploaded PDF**. As shown in the table, our method is quite robust to different values of these hyperparameters (e.g., α, β). We have updated the manuscript with the settings and specific values of these parameters.
>
> 2) The use of multiple multi-feature extraction blocks has proven to be effective in enhancing performance [1]. The results in the following table also indicate that increasing the number of these blocks leads to further performance improvements. However, considering computational costs, we have decided to utilize four blocks in our final implementation to achieve an optimal balance between performance and efficiency.
>
> | Number |   2   |   3   |   4   |   5   |
> |:------:|:-----:|:-----:|:-----:|:-----:|
> | AUROC  | 95.7  | 96.0  | 96.2  | 96.4  |
> ***
>
> ***
>
> >W#3: The equations are not numbered and the line numbers are incomplete which makes it hard to reference.
>
> **A#3:** Thank you for your helpful suggestion. we will incorporate the modifications in the version.
>
> ***
>
> ***
>
> >Q#1: What are the exact components of C0 and C0? As for the normal state prompts, is the number of n below Line125 13? And what is the number of the memory bank ?
>
>
> **A#4:** 1)   We follow the notation used in the DreamBooth method to customize a diffusion model. ${C_0}$ consists of a set of text-image pairs, $ C_0 = \{(x_k, c_k)\} $, centered around the target object. The components of ${C_0}$ are reference images (i.e., few-shot normal samples) of the same object (e.g., cable, candle, brain, etc.) and their corresponding prompts. Additionally, $\overline{C_o}$ contains different images (i.e., regularization images) within the same object for prior preservation and regularization purposes. The regularization images are generated using the Stable Diffusion (SD) model, as is common in most methods. These images use a prompt that is a coarse class descriptor to prevent language drift and reduce output diversity. 2) The number \( n \) in our method is 3 and the number of memory bank is 30.
>
> ***
> ***
>
> >Q#2: The "Triplet Contrastive Anomaly Inference" seems to work as a weighted prediction from three comparison aspects during testing. Where can it reflect the concept of "contrastive"? What is the full training objective?.
>
> **A#5:** Thank you for your detailed question on the "Triplet Contrastive Anomaly Inference" part. Yes, the "Triplet Contrastive Anomaly Inference" method we propose involves a weighted prediction from three comparative aspects during the testing phase. The term "contrastive" in our context refers to the dissimilarity comparisons among three branches: the query image in comparison with the personalized image, anomaly-free samples, and text prompts. This is designed not to interfere with the training process.
>
> ***
> ***
>
> >Q#3: Does reflect the degree of anomaly score? It seems that there are two types (both normal and abnormal) objects in the text prompts. How do they work in the same way as the equations of  and ?
>
>
> **A#6:** Indeed, in our method, text prompts contain two types of objects: normal and abnormal. During the computation of $S_{\text{text}}$, the query image is compared with prompts of both types, yielding two probabilities: $p_{\text{normal}}$ and $p_{\text{abnormal}}$, which represent the likelihood of the image being normal and abnormal, respectively. Subsequently, the anomaly score $S_{\text{text}}$ is calculated by summing $p_{\text{abnormal}}$ and $1 - p_{\text{normal}}$, which is then utilized for subsequent calculations of $A_{\text{score}}$.
>
>
> ***
>
> ***
>
> >Q#4: Is there any limitation for the proposed method to handle the open-vocabulary scenarios? And what is the computation cost and inference cost of the proposed method compared with other methods since there are many images for generation?
>
> **A#7:** Thank you for your good suggestion. 1) Our method relies on learning a representative distribution of normal samples and therefore has to see few-shot normal examples.  We have not yet considered the direction of open-vocabulary scenarios. This could be a potential area for further exploration in future work. 2）Please refer to A#1.
>
>
>
> ***
>
> [1] Toward Generalist Anomaly Detection via In-context Residual Learning with Few-shot Sample Prompts, CVPR 2024

---

### Official Review · Reviewer_Ljby · 2024-07-11

**Soundness:** 3
**Presentation:** 4
**Contribution:** 2
**Rating:** 5
**Confidence:** 5

**Summary:**

The paper focuses on a practical yet challenging anomaly detection in a few-shot-normal-image setting. Instead of directly matching features between the query image and a few normal reference images, the core insight is to replace the reference image with a personalized normal image generated by an anomaly-free custom model. The authors also propose a triplet contrastive anomaly inference strategy, incorporating the original/generated anomaly-free samples and text prompts. Extensive experiments are extensively conducted across 11 datasets.

**Strengths:**

1. The paper is well-writing and easy to follow.
2. State-of-the-art results are achieved across 11 datasets.

**Weaknesses:**

1. It is essentially a memory-augmented reconstruction-based anomaly detection (AD) method [1], which attempts to reconstruct the query image to its most similar anomaly-free counterpart. However, the reconstruction-based AD method also explores the Stable Diffusion (SD) denoising network [1]. Could you clarify the differences? If I understand correctly, the core difference is the inputs to SD, which are pairs of object text prompts and few-shot normal images.
2. Though Figure 4 demonstrates the effectiveness of the generated anomaly-free samples on three AD methods, how do these generated samples enhance the authors’ method? Can we use more or fewer generated normal samples instead of 100? An ablation study is required here.

[1] Gong et al. Memorizing normality to detect anomaly: memory-augmented deep autoencoder for unsupervised anomaly detection. In ICCV, 2019.
[2] He et al. Diad: A diffusion-based framework for multi-class anomaly detection. In AAAI, 2024.

**Questions:**

1. Which data augmentations are used to augment few-shot normal images? Do these data augmentations vary across different categories?
2. It would be interesting to investigate how text prompts affect the anomaly-free customized model. For example, what if a specific category name (e.g., “cable”) is used to replace "object"?
3. How are \alpha and \beta determined in the final prediction? How about the ratio of the t-step?
4. It would be favorable to report F1-max, AP and PRO along with AUROC.
5. What about the one-shot setting?

**Limitations:**

The limitations are adequately addressed.

---

> ### Author Rebuttal · Authors · 2024-08-07
>
> We thank the reviewer for the careful reviews and constructive suggestions. We answer the questions as follows.
> ***
> >W#1: It is essentially a memory-augmented reconstruction-based anomaly detection (AD) method [1], which attempts to reconstruct the query image to its most similar anomaly-free counterpart. However, the reconstruction-based AD method also explores the Stable Diffusion (SD) denoising network [1]. Could you clarify the differences? If I understand correctly, the core difference is the inputs to SD, which are pairs of object text prompts and few-shot normal images.
>
> **A#1:** Thanks for your constructive question. Indeed, our approach and memory-augmented reconstruction-based anomaly detection both attempt to reconstruct the query image to its most similar anomaly-free counterpart. There are some core differences: (1) Our reconstruction process does not utilize a memory bank. As the core differences you mentioned, It relies on Stable Diffusion (SD) using pairs of object text prompts and few-shot normal images. (2) Their memory-augmented reconstruction-based anomaly detection is entirely based on memory bank features for reconstruction, without any feature information from the original input image. In such a scenario, the most similar features found in the memory bank may still have low similarity because no information from the original input image is used for reconstruction, and the features stored in the memory bank may not fully match the input image, which may lead to increased reconstruction errors or noise. (3) Instead, We explore a stable diffusion process that allows control over the generative process and retains most of the original image information by converting anomalous regions of the query image to normal, thereby reconstructing the query image to its most similar anomaly-free counterpart. This method preserves most of the normal regions of the original image in the reconstructed image, reducing errors. (4) The reconstruction-based anomaly detection methods in [1] and [2] detect anomalies entirely by computing the difference between the original and reconstructed images, whereas our personalized image reconstruction is only one of the three branches used to compute the anomaly score.
> ***
> ***
> >W#2: Figure 4 demonstrates the effectiveness of the generated anomaly-free samples on three AD methods, how do these enhance the authors’ method? Can we use more or fewer generated normal samples instead of 100?
>
> **A#2:** Thank you for the detailed question. 1) The generated samples are involved in calculating the anomaly score, which is discussed in Triplet Contrastive Anomaly Inference section. These normal samples are generated to compare multi-level features with query images, contributing to one aspect of the anomaly score, denoted as $S_{N}$. Additionally, the ablation study in Table 5 of the manuscript confirms that these generated samples enhance the performance.  2) Yes, we attempted to validate the effectiveness of the proposed method by generating samples ranging from 10 to 300. For your convenience, we have placed the results in Figure 1 of **the uploaded PDF**. Increasing generated samples continues to enhance the results, likely because a larger sample set better represents the distribution of normal samples. However, for computational efficiency, we do not recommend generating too many samples (e.g., exceeding 200).
> ***
> ***
> >Q#1: Which data augmentations are used to augment few-shot normal images?Do these data augmentations vary across different categories?
>
> **A#3:** Thank you for the detailed question. We performed only simple data augmentation, such as flipping and Gaussian blur, on medical datasets as well as the KSDD dataset. This approach was adopted primarily to preserve the distribution of normal samples. Additionally, our method has a comprehensive text prompts design which, unlike data augmentation, can more effectively simulate various states of normal images.
> ***
> ***
> >Q#2: It would be interesting to investigate how text prompts affect the anomaly-free customized model. For example, what if a specific category name (e.g., “cable”) is used to replace "object"?
>
> **A#4:** Thanks for your good suggestion. In the experiments, we indeed used the specific category name, e.g., “cable”  to replace the variable “object”. This is to be consistent with recent related work using textual prompts. To avoid confusion, we will clarify this in our revised manuscript.
> ***
> ***
> >Q#3: How are alpha and beta determined in the final prediction? How about the ratio of the t-step?
>
> **A#5:** Thank you for the detailed question.1) We set the parameters $\alpha$ and $\beta$ for $A_{score}$ to 1 and 0.5, respectively, this configuration remains consistent across all datasets. We provide Table 1 in **the uploaded PDF**, demonstrating the robustness of our method to different choices of hyperparameters.
> 2）The ratio for the t-step is set at 0.3. In experiments, we observed that performance fluctuations remain minimal within the range of 0.2 to 0.5. A ratio of 0.3 not only performs well across the majority of datasets but also offers enhanced computational efficiency due to its smaller value.
> ***
> ***
> >Q#4: It would be favorable to report F1-max, AP and PRO along with AUROC？
>
> **A#6:** Indeed, presenting metrics such as maximum F1-max,  AP, and probability of PRO, along with AUROC, would be the most favorable and comprehensive approach. However, given the breadth of our data—comprising 11 datasets across three domains—and the limited page space, it is not feasible to display all these indicators. Therefore, similar to previous methods like InCTRL and RegAD, we prioritize presenting AUROC.
> ***
> ***
> >Q#5: one-shot setting
>
> **A#7:** Thanks for your constructive suggestions. We have calculated the results of our method and compared them with other methods in a one-shot setting. For your convenience, these results are presented in Table 2 in **the uploaded PDF**.
> ***

---

> ### Author Response · Authors · 2024-08-08
>
> For your convenience, we provide the experimental results mentioned in **the uploaded PDF** here to facilitate your review. We hope these supporting data address any concerns or questions you may have. If there is any confusion or if any part of our work requires further clarification, please do not hesitate to comment. We are more than willing to provide additional explanations and engage in further discussions.
>
> ***
> >W#2: Can we use more or fewer generated normal samples instead of 100? An ablation study is required here.
>
> **A#1:** Yes, we attempted to generate samples ranging from 10 to 300 to validate the effectiveness of the proposed method. We have placed the results here in the following table. Increasing the number of generated samples continues to enhance the results, likely because a larger sample set better represents the distribution of normal samples. However, for computational efficiency, we do not recommend generating too many samples (e.g., exceeding 200).
>
> | Datasets | 10   | 30   | 50   | 100  | 150  | 200  | 300  |
> |----------|------|------|------|------|------|------|------|
> | MVTec-AD | 95.9 | 96.2 | 96.3 | 96.4 | 96.5 | 96.6 | 96.6 |
> | RESC     | 94.7 | 95.2 | 95.4 | 95.6 | 95.6 | 95.7 | 95.9 |
> | CIFAR    | 94.2 | 94.9 | 95.2 | 95.5 | 95.6 | 95.6 | 95.7 |
>
> ***
>
> ***
> >Q#3: How are alpha and beta determined in the final prediction?
>
> **A#2:** We set the parameters $\alpha$ and $\beta$ for $A_{score}$ to 1 and 0.5, respectively, this configuration remains consistent across all datasets. This choice was informed by our preliminary experiments, which demonstrated satisfactory performance across the majority of datasets under this setting. We provide more results in the following table below, demonstrating the robustness of our method to different choices of $\alpha$ and $\beta$.
>
> | $\alpha$ | $\beta$ | MVTec | VisA | KSDD | AFID | ELPV | OCT2017 | BrainMRI | HeadCT | RESC | MNIST | CIFAR-10 | Average|
> |:--------:|:-------:|:-----:|:----:|:----:|:----:|:----:|:-------:|:--------:|:------:|:----:|:-----:|:--------:|:--------:|
> |    1     |   0.5   | 96.2  | 89.9 | 98.4 | 84.7 | 90.6 |  99.3   |   98.6   |  94.8  | 95.2 | 93.6  |   94.9   |   **94.2**   |
> |   0.5    |    1    | 95.7  | 89.3 | 98.0 | 84.2 | 90.1 |  99.1   |   98.3   |  94.6  | 95.1 | 93.2  |   94.6   |   **93.9**   |
> |    1     |    1    | 95.9  | 89.5 | 98.1 | 84.1 | 90.2 |  99.4   |   98.5   |  95.1  | 95.0 | 93.4  |   94.7   |   **94.0**   |
> |   0.5    |   0.5   | 96.1  | 89.7 | 98.3 | 84.6 | 90.8 |  99.0   |   98.1   |  94.5  | 95.3 | 93.8  |   94.3   |   **94.1**   |
> |    2     |    1    | 96.0  | 89.8 | 98.5 | 84.0 | 90.0 |  99.2   |   98.3   |  94.3  | 94.8 | 93.2  |   94.6   |   **93.9**   |
> |    1     |    2    | 95.7  | 89.1 | 97.8 | 83.5 | 89.7 |  99.2   |   98.7   |  94.5  | 95.0 | 93.4  |   94.7   |   **93.8**   |
> |    2     |    2    | 95.9  | 89.3 | 97.9 | 83.2 | 89.9 |  99.1   |   98.5   |  95.3  | 94.8 | 93.1  |   94.8   |   **93.8**   |
>
>
> ***
>
> ***
>
> >Q#5: What about the one-shot setting?
>
> **A#3:** In the one-shot setting, we have calculated the performance of our method and compared it with recent methods. The table below presents the AUROC comparison results, showing that our method maintains optimal performance on most datasets.
>
> |                   | Datasets        | WinCLIP                   | InCTRL                   | Ours                       |
> |:-----------------:|:---------------:|:-------------------------:|:------------------------:|:--------------------------:|
> | **Industrial field** | **MVTec**       | 92.5±2.3                  | 93.2±1.7                 | **94.8±0.7**               |
> |                   | **VisA**        | 83.6±2.5                  | 84.2±2.5                 | **87.0±1.7**               |
> |                   | **KSDD**        | 94.0±0.5                  | **96.6±2.8**             | 96.5±1.6                   |
> |                   | **AFID**        | 72.3±4.2                  | 76.0±3.2                 | **77.6±1.5**               |
> |                   | **ELPV**        | 72.2±2.5                  | 82.8±1.2                 | **85.2±0.8**               |
> | **Medical field**    | **OCT2017**     | 90.7±2.6                  | 93.0±2.3                 | **95.8±1.6**               |
> |                   | **BrainMRI**    | 93.1±1.5                  | 96.7±2.4                 | **96.9±1.3**               |
> |                   | **HeadCT**      | 91.7±1.8                  | 92.3±2.0                 | **93.7±1.2**               |
> |                   | **RESC**        | 85.7±2.6                  | 87.6±2.9                 | **92.4±1.2**               |
> | **Semantic field** | **MNIST**       | 76.3±1.7                  | 87.7±2.3                 | **91.8±0.6**               |
> |                   | **CIFAR-10**    | 92.3±0.2                  | 93.2±0.9                 | **93.6±0.5**               |
>
> ***

---

> ### Author Response · Authors · 2024-08-08
>
> ***
>
> >Q#4: It would be favorable to report F1-max, AP and PRO along with AUROC？
>
> **A#4:** Following your suggestion, we have included the results of AUROC, F1-max,  AP, AUPRC, and PRO on MVTec dataset in the table below for a more comprehensive comparison, where our method consistently outperforms the baselines on all metrics.
>
>
> |                |  Methods  |  AUROC  |  F1-max  |   AP   |  AUPRC  |  PRO   |
> |:--------------:|:---------:|:-------:|:--------:|:------:|:-------:|:------:|
> | **2-shot**     | WinCLIP   |  93.1   |   93.3   |  95.9  |  96.5   |  88.2  |
> |                | InCTRL    |  94.0   |    -     |   -    |  96.9   |   -    |
> |                | VAND      |  92.4   |   92.6   |   96   |    -    |  91.3  |
> |                | Ours      | **95.1**| **94.3** |**96.5**|**97.3** |**92.1**|
> | **4-shot**     | WinCLIP   |  94.0   |   93.5   |  96.2  |  96.8   |  88.5  |
> |                | InCTRL    |  94.5   |  **-**   |   -    |  97.2   |   -    |
> |                | VAND      |  92.8   |   92.8   |  96.3  |    -    |  91.8  |
> |                | Ours      | **95.6**| **94.8** | **97** |**97.8** |**92.6**|
> | **8-shot**     | WinCLIP   |  94.7   |   93.8   |  96.5  |  95.3   |  89.1  |
> |                | InCTRL    |  95.3   |    -     |   -    |  97.7   |   -    |
> |                | VAND      |  93.0   |   93.1   |  96.5  |    -    |  92.2  |
> |                | Ours      | **96.2**| **95.1** |**97.4**|**98.9** |**93.1**|
>
>
> ***

---

> ### Comment · Reviewer_Ljby · 2024-08-11
>
> I highly appreciate the authors' helpful feedback. I agree with explanations on the differences between the diffusion model-based reconstruction and memory-based one. All other questions are well-solved through additional experiments so I would like to increase my rating and recommend a borderline acceptance.

---

> > ### Author Response · Authors · 2024-08-12
> >
> > We sincerely appreciate you taking the time to review our rebuttal and for your positive feedback. We are very glad that our response has addressed your concerns.

---

### Author Rebuttal · Authors · 2024-08-07

We appreciate all reviewers for their careful reviews and constructive suggestions. In this rebuttal, Individual concerns have been carefully addressed in the response to each reviewer, with an uploaded PDF for more results suggested by reviewers. In the final version, we will revise the paper following these suggestions.

---

### Decision · Program_Chairs · 2024-09-25

**Decision:**

Accept (poster)

**Comment:**

All reviewers are positive about this paper. They acknowledge the novelty of this paper and the achieved superior performance. They also have some questions about the ablation study and other analysis issues, which are well-addressed during the rebuttal. Therefore, the AC agrees with the positive opinion of the reviewers and recommends acceptance.